# New Insights into the Genetic Basis of Lysine Accumulation in Rice Revealed by Multi-Model GWAS

**DOI:** 10.3390/ijms25094667

**Published:** 2024-04-25

**Authors:** Liqiang He, Yao Sui, Yanru Che, Lihua Liu, Shuo Liu, Xiaobing Wang, Guangping Cao

**Affiliations:** 1School of Tropical Agriculture and Forestry, Hainan University, Haikou 570228, China; 2Institute of Tropical Crop Genetic Resources, Chinese Academy of Tropical Agricultural Sciences, Danzhou 571737, China; 3Hainan Key Laboratory of Crop Genetics and Breeding, Institute of Food Crops, Hainan Academy of Agricultural Sciences, Haikou 571100, China

**Keywords:** rice, free lysine content, genome-wide association study, quantitative trait nucleotide, gene, QTN-by-environment interaction, genomic selection/prediction

## Abstract

Lysine is an essential amino acid that cannot be synthesized in humans. Rice is a global staple food for humans but has a rather low lysine content. Identification of the quantitative trait nucleotides (QTNs) and genes underlying lysine content is crucial to increase lysine accumulation. In this study, five grain and three leaf lysine content datasets and 4,630,367 single nucleotide polymorphisms (SNPs) of 387 rice accessions were used to perform a genome-wide association study (GWAS) by ten statistical models. A total of 248 and 71 common QTNs associated with grain/leaf lysine content were identified. The accuracy of genomic selection/prediction RR-BLUP models was up to 0.85, and the significant correlation between the number of favorable alleles per accession and lysine content was up to 0.71, which validated the reliability and additive effects of these QTNs. Several key genes were uncovered for fine-tuning lysine accumulation. Additionally, 20 and 30 QTN-by-environment interactions (QEIs) were detected in grains/leaves. The QEI-sf0111954416 candidate gene *LOC_Os01g21380* putatively accounted for gene-by-environment interaction was identified in grains. These findings suggested the application of multi-model GWAS facilitates a better understanding of lysine accumulation in rice. The identified QTNs and genes hold the potential for lysine-rich rice with a normal phenotype.

## 1. Introduction

Lysine is one of the nine essential amino acids (EAAs), which cannot be synthesized in humans and needs to be obtained from external diets, especially from plant-based diets [1,2,3,4]. Lysine in the human diet comes from the digestion of lysine-containing proteins rather than from free lysine in the plant or animal cell. Lysine deficiency in the human body leads to health concerns such as retarded growth, tiredness, anemia, calcium absorption, chronic malnutrition, and antibody production [2,5]. Rice represents an important staple food, which is a major source of calories and amino acids intake for humans and livestock [6,7,8]. However, the inadequate nutritional value of rice is the EAA lysine, which is known as the most limiting amino acid and a limitation of the nutrient quality [2,5,9]. Thus, enhancing the lysine content in rice is becoming an emerging goal to meet the nutrition demands of the ever-growing global population.

In order to increase the rice lysine accumulation in a feasible and cost-effective manner, extensive efforts on biofortification have been made using genetic and metabolic engineering strategies [1,2,10]. Most of this research concentrated on enhancing lysine anabolism and reducing lysine catabolism [3,4,11]. To date, the characterized mechanism of lysine biosynthesis and degradation remains far from comprehensive and detailed. Lysine is synthesized through a branch of the aspartate (Asp) family pathway; two key enzymes involved in the lysine biosynthesis are Asp kinase (AK) and dihydrodipicolinate synthase (DHDPS), and one important enzyme participating in the lysine degradation pathway is the bifunctional lysine ketoglutarate reductase/saccharopine dehydrogenase (LKR/SDH) [3,4,12]. For instance, the expression of bacterial *DHDPS* with a seed-specific promoter in an *LKR/SDH* knockdown mutant line showed an approximately 64-fold increase of free lysine content in Arabidopsis [13,14]. The overexpression of maize *DHDPS* in rice results in only a 2.5-fold increase of lysine in mature grains but with a low seed germination rate [15]. Using the combined expression of *AK* and *DHPS* with RNA interference of the *LKR/SDH* approach in rice, the free lysine contents increased up to 12-fold in leaves and 60-fold in seeds [16]. However, the strengths and constraints are always presented in these transgenic lines with relatively high lysine levels, which are generally accompanied by some deleterious effects, such as alterations in plant height, germination rate, seed vigor, seed color, and oil content [10,17,18]. Therefore, further study is likely needed to fully elucidate the genetic mechanism for a comprehensive understanding of metabolic fluxes about the lysine-related pathway.

The genome-wide association study (GWAS) detects the marker and trait associations in a powerful and robust manner, which is commonly used in the genetic research of the quantitative trait controlled by polygenes [19,20,21,22]. Due to the differences in the genetic algorithm, GWAS models can be mainly classified into the generalized linear model (GLM), mixed linear model (MLM), and its derived single-locus models (MLM, GEMMA, EMMAX, and CMLM etc.), and multi-locus models (MLMM, FarmCPU, mrMLM, pLARmEB, FASTmrEMMA, pKWmEB, FASTmrMLM, ISIS EM-BLASSO, and 3VmrMLM etc.) [23,24,25,26,27,28,29,30,31,32,33]. These models have several advantages and also a few shortcomings. For example, MLM and its derived single-locus models have been proven to control spurious associations and show high performance in the detection of quantitative trait nucleotide/locus (QTN/QTL) with a large effect [30,34,35]. Thus, a range of multi-locus models have been proposed to detect the QTNs/QTLs with large and small effects in an accurate and robust manner [35,36,37,38]. According to the results obtained in our previous studies, the combined use of multiple statistical models for GWAS facilitates a better understanding of the genetic mechanism of the complex and multi-omics trait, particularly for the free amino acid content in rice [36,37,38,39].

To date, the content alteration is well explained by the genetic variation through the metabolite-based genome-wide association study (mGWAS), which provides refined mechanistic insights into primary and secondary metabolic genes and pathways in plants. To unravel the genetic basis underlying the primary and secondary metabolic interface, various QTLs/QTNs and candidate genes have been exploited using this approach [37,38,40,41,42,43,44,45,46,47]. Previous studies generally focused on the structural genes in the biosynthesis and metabolism pathways. However, transcription factors (TFs) are powerful tools for regulating the biosynthesis and degradation of certain metabolites, which activate or suppress the expressions of multiple genes participating in one or more pathways [48,49]. As a primary metabolite, the accumulation of a free branched-chain amino acid in rice has been uncovered by the positive regulation of the TF *OsWRKY78* in grains and *OsbZIP18* in leaves [45,46]. In contrast, the theanine (a non-proteinaceous amino acid) biosynthesis in tea is negatively regulated by the TF *CsMYB73*, and the proanthocyanidin accumulation in grape berry is also negatively regulated by the TF *VvMYBC2-L1* [50,51]. The accumulation of free lysine in rice is mainly related to the biosynthesis in leaves and catabolism in seeds, and no evidence shows that free lysine transports from leaves into seeds [16]. Therefore, the exploration of key genes and corresponding TFs in rice grains and leaves is of utmost importance for the fine-tuning of the lysine biosynthesis and metabolism pathways. Additionally, to confront the challenge of global climate change and meet the nutrition demands of the increasing population, the genetic basis of the QTN-by-environment interactions (QEIs) related to the lysine content in rice grains and leaves needs to be elucidated.

To explore the QTNs and potential genes enhancing lysine accumulation in rice grains and leaves, a GWAS was performed on a diverse panel of 387 rice accessions with 4,630,367 SNPs. This association panel contained 244 *indica* accessions and 143 *japonica* accessions. The QTNs and QEIs associated with the lysine content in rice grains and leaves were detected across five grain and three leaf lysine content datasets using multiple GWAS models. These models included GLM, three single-locus models (MLM, CMLM, and EMMAX), and six multi-locus models (mrMLM, FASTmrEMMA, FASTmrMLM, ISIS EM-BLASSO, pKWmEB, and pLARmEB). The objective of this study was to identify the QTNs, key genes contributing to lysine accumulation, and the QEIs related to the lysine content in rice.

## 2. Results

### 2.1. Lysine Content in Rice Grains and Leaves

To assess the content variation of lysine in 387 rice accessions, LC-MS/MS technology was used to quantify the lysine content in grains and leaves. The coefficient of variation (CV) of lysine content in grains ranged from 93.59% to 165.98%, whereas in leaves, it ranged from 43.53% to 52.62% (Table 1). The skewness and kurtosis of all the lysine content datasets were observed in less than one (Table 1). The estimated broad-sense heritability (*H*^2^) of grain/leaf lysine content was 0.69 and 0.16 (Table 1). Interestingly, significant differences in grain/leaf lysine content were observed between *japonica* and *indica* rice. Higher grain lysine levels were found in *indica* accessions than in *japonica* accessions (Figure 1A). However, higher leaf lysine contents were observed in *japonica* accessions than in *indica* accessions (Figure 1C). Correlation analyses were conducted among five grain lysine content datasets (Grain_env1_r1, Grain_env1_r2, Grain_env2_r1, Grain_env2_r2, and Grain_BLUP) and three leaf content datasets (Leaf_env3_r1, Leaf_env3_r2, and Leaf_BLUP). Across all the grain lysine content datasets, the highest correlation coefficients (*r* = 0.94) were observed between Grain_env1_r2 and Grain_BLUP and between Grain_env2_r2 and Grain_BLUP, while the lowest was found between Grain_env1_r1 and Grain_env2_r1 (*r* = 0.72) (Figure 1B). In the leaf lysine content datasets, the highest correlation relationship was found between Leaf_env3_r2 and Leaf_BLUP (*r* = 0.86), while the lowest was observed between Leaf_env3_r1 and Leaf_env3_r2 (*r* = 0.40) (Figure 1D). The data distribution and correlation results indicated the lysine content in rice grains and leaves is quantitatively inherited and affected by genetic and environmental interactions.

### 2.2. Population Analysis

To analyze the genetic structure of the 387 rice accessions, the identified 107,761 SNPs were used for the assessment of the genetic relationship. These accessions were divided into two groups and comprised of 244 *indica* accessions and 143 *japonica* accessions by the principal component analysis (PCA) (Figure 2A,B). The consistent classification was obtained by the population structure and neighbor-joining (NJ) tree-based phylogenetic analyses (Figure 2C,D). Therefore, a population structure matrix with K = 2 was used for the subsequent GWAS analyses. The *r*^2^-based linkage disequilibrium (LD) analysis showed the averaged whole genome LD of this genetic panel was approximately 122 kb. Additionally, a higher decay rate was observed in *indica* accessions than that in *japonica* accessions (Figure 2E). Therefore, the 122 kb flanking region of each QTN was used for putative candidate gene prediction in the following analyses.

### 2.3. Identification and Application of QTNs Associated with Lysine Content

Using ten statistical models, a total of 43,569 and 29,115 putative QTNs were detected on the basis of 387 rice accessions with 4,630,367 SNPs and eight content datasets (Grain_env1_r1, Grain_env1_r2, Grain_env2_r1, Grain_env2_r2, and Grain_BLUP for the grain lysine content, Leaf_env3_r1, Leaf_env3_r2, and Leaf_BLUP for the leaf lysine content) (Appendix A). According to the differences in these genetic algorithms, ten GWAS models were classified into GLM, MLM-based single-locus model (MLM-SL), and mrMLM-series multi-locus model (mrMLM-ML) for the further identification of common QTNs of each lysine dataset. MLM-SL contained MLM, CMLM, and EMMAX models. mrMLM-ML included mrMLM, FASTmrEMMA, FASTmrMLM, ISIS EM-BLASSO, pKWmEB, and pLARmEB. The largest number of QTNs was detected by the GLM model in each lysine content dataset, while the smallest number of QTNs was generally detected by MLM-SL and mrMLM-ML. For example, no QTN was detected by the FASTmrEMMA and mrMLM models in Leaf_BLUP (Appendix A). A QTN detected by two or more statistical models in a lysine dataset was defined as a common QTN. In total, 248 and 71 common QTNs were identified as potentially underlying the grain/leaf lysine content in rice (Appendix A). The largest number of common QTNs associated with the grain/leaf lysine content were detected in Grain_BLUP (117 QTNs) and Leaf_env3_r2 (30 QTNs) (Table 2). The phenotypic variance explained (PVE) by the common QTN associated with grain lysine content ranged from 0.03% to 27.65%, and the PVE by the common QTN associated with leaf lysine content was from 0.03% to 16.18% (Table 2 and Appendix A). A vast majority of common QTNs were detected by the GLM model in all lysine content datasets, except for 20 and 8 common QTNs co-detected by MLM-SL and mrMLM-ML models in the grain/leaf lysine content dataset (Appendix A). Moreover, the combined use of GLM and MLM-SL models identified the highest number of common QTNs (164) in grain lysine datasets (Table 2). For instance, QTN-sf0825353310 was co-detected by GLM, MLM, CMLM, EMMAX, FASTmrMLM, ISIS EM-BLASSO, mrMLM, and pLARmEB in Grain_BLUP dataset (Appendix A). However, the most common QTNs (38) in leaf lysine datasets were identified by the GLM and mrMLM-ML models (Table 2), such as QTN-sf0702729577 detected by GLM, MLM, CMLM, EMMAX, FASTmrEMMA, FASTmrMLM, ISIS EM-BLASSO, mrMLM, and pLARmEB in the Leaf_env3_r2 dataset (Appendix A). The position and number of all detected putative and common QTNs associated with the lysine content in rice grain/leaf are shown on a CIRCOS map (Appendix A). Furthermore, ten and six common QTNs were co-detected in more than two lysine content datasets in grains/leaves separately, which were considered as the dataset stable QTNs (Appendix A).

To assess the potentials of these QTNs for nutrient quality breeding, 16 genomic selection/prediction (GS/GP) models were constructed on the basis of 248 and 71 common QTNs and eight lysine content datasets (five for grain lysine, and three for leaf lysine). Using the five-fold cross-validation scheme, the highest predictive ability was generated in the Grain_BLUP and Leaf_BLUP datasets with the accuracy of (*r*) 0.85 and 0.77 (Table 3). Correspondingly, their SNP-based heritability (*h*^2^) was estimated to be up to 0.64 and 0.34 (Table 3). In addition, the dataset stable QTNs and the grain/leaf content datasets were used to test the additive effect further. The significant correlation between the number of favorable alleles (NFA) and the lysine content ranged from 0.59 to 0.71 in grains and 0.43 to 0.51 in leaves (Appendix A).

### 2.4. Candidate Genes for the Lysine Accumulation in Rice Grains

For the prediction of the genes putatively underlying the grain lysine content in rice, a total of 3550 genes were identified (Appendix A). To uncover the key genes involved in the amino acid-related pathways, a KEGG pathway analysis was conducted, which showed several genes played important roles in the lysine biosynthesis, lysine degradation, biosynthesis of amino acids, alanine, aspartate and glutamate metabolism, beta-Alanine metabolism, cyanoamino acid metabolism, cysteine and methionine metabolism, and tryptophan metabolism pathways (Figure 3 and Table 4). Of these genes, the lysine biosynthesis gene *LOC_Os07g20544* encoding aspartokinase (AK) protein was localized in the LD block Chr7: 11,864,886–11,951,886 bp of QTN-sf0711949886 (sf0711949886 indicates chromosome 7 at 11,949,886 bp) locus (Figure 4A and Table 4). Furthermore, the haplotypic variation of it was examined. The *AK* gene carried three haplotypes which included Hap1 (GGCCGGAATTTTGG, *n* = 314), Hap2 (CCCCAACCCCCCAA, *n* = 41), and Hap3 (GGAAGGAATTTTGG, *n* = 27) (Figure 4B). Across all the grain lysine content datasets, significantly lower lysine levels were observed in the accessions with Hap2 compared to accessions carrying Hap1 and Hap3 (Figure 4C–F). To further explore the potential regulators of this *AK* gene, the transcript factor (TF) binding site analysis was performed using the web tool PlantRegMap (Table 4). Interestingly, a TF gene *LOC_Os12g32250* (WRKY DNA-binding domain-containing protein, namely WRKY) binding to the cis-elements in the *AK* gene promoter region with the matched motif sequence CCTAGTCAACC was also a candidate gene of another QTN-sf1219521482 locus (Figure 4G, Table 4, and Appendix A). Similarly, haplotype and content analysis of this *WRKY* TF showed significantly higher lysine contents in the accessions with Hap1 (GGTT, *n* = 254) than in the accessions carrying Hap2 (AACC, *n* = 131) (Figure 4H–L). For the subsequent investigation of the expression profile of the *WRKY* TF and *AK* gene, the seed and leaf RNA-seq data of the *japonica* rice Nipponbare, the *indica* rice Minghui63, and Zhenshan97 were used. A similar expression pattern was observed between the *WRKY* TF and *AK* gene in Nipponbare (*r* = 0.90), Minghui63 (*r* = 0.25), and Zhenshan97 (*r* = 0.45) varieties (Appendix A).

### 2.5. Candidate Genes for the Lysine Accumulation in Rice Leaves

In order to identify the putative genes associated with lysine content in rice leaves, a number of 1893 genes localized within the flanking region based on the averaged whole genome LD decay. Subsequently, a KEGG pathway analysis showed these genes mainly enriched in the amino acid accumulation and metabolism-related pathways, such as lysine degradation, biosynthesis of amino acid, alanine, aspartate, and glutamate metabolism, and cysteine and methionine metabolism (Figure 3 and Table 5). Of note, the amino acid biosynthesis gene *LOC_Os11g33240*, localized in the LD block Chr11: 19,081,279–19,190,279 bp, encoding citrate synthase (CS) enzyme was a candidate gene harboring in the QTN-sf1119083279 locus (Figure 5A and Table 5). The haplotypic variation analysis showed this *CS* gene had seven functional haplotypes, which contained Hap1 (GGCCGGAA, *n* = 212), Hap2 (GGTTGGAA, *n* = 82), Hap3 (TTCCAAAA, *n* = 45), Hap4 (GGCCGGTT, *n* = 41), Hap5 (GTCCAAAA, *n* = 3), Hap6 (GGCCGGAT, *n* = 2), and Hap7 (GTCCGAAA, *n* = 2) (Figure 5B). Significant higher leaf lysine levels were observed in the accessions with Hap2 compared with those accessions carrying the other Haps in all the leaf lysine content datasets (Figure 5C,D). Furthermore, a *MYB* TF (*LOC_Os01g19970*) binding to the cis-elements in the *CS* gene promoter region (matched motif sequence: CAACCTACCG) was predicted by the web tool PlantRegMap (Table 5). This *MYB* TF localized in the LD block of Chr1: 11,238,543–11,356,543 bp was a candidate gene of another QTN-sf0111240543 locus (Figure 5E, Table 5, and Appendix A). Additionally, the accessions carrying Hap2 (GGGGCC, *n* = 153) exhibited a significantly higher lysine level in leaves than those with Hap1 (GGTTCC, *n* = 185) of the *MYB* TF in the Leaf_env3_r1 dataset (Figure 5G). However, no significance of the leaf lysine content was shown among the accessions with the three haplotypes in the Leaf_env3_r2 dataset (Figure 5H). In Nipponbare, a similar expression trend between the *MYB* TF and *CS* gene was shown in Appendix A (*r* = 0.90).

### 2.6. Candidate Regulators Underlying the Lysine Accumulation in Rice Grains and Leaves

Notably, two transcription factors (TFs) potentially regulating the expression of *AK* and *CS* genes were identified in the candidate genes associated with the grain/leaf content in rice. For example, using the TF binding site prediction of PlantRegMap, the *AK* and *CS* gene promoter regions both contained the matched cis-element sequence TTCTTTTCTATTTTATAAA of the TF *LOC_Os01g10504* (MADS-box family gene with MIKCc type-box, MADS). This *MADS* TF localized in the LD block of Chr1: 5,537,291–5,568,291 bp, which was also a candidate gene of the QTN-sf0105539291 locus associated with the grain lysine content (Figure 6A). Moreover, a relatively high lysine content of the accessions with the functional haplotype Hap2 (CC, *n* = 163) of the *MADS* TF was observed than those with Hap1 across all the grain lysine content datasets (Figure 6B–F). On the basis of the expression data in Nipponbare, an almost opposite expression pattern of the *MADS* TF and the *AK* gene was observed (Figure 6G). Correlation analyses showed the correlation coefficient between the *MADS* and *AK* was −0.60, while the correlation coefficient between the *MADS* and *CS* was 0.10. In Minghui63, the different expression patterns of the *MADS* TF, *AK*, and CS genes are shown in Figure 6H (*r* = −0.42 between *MADS* and *AK*, *r* = −0.30 between the *MADS* and *CS*). Consistent with the results obtained in Minghui63, a distinct expression profile of the *MADS* TF compared to the *AK* and CS genes was observed in Zhenshan97 (*r* = −0.63 between *MADS* and *AK*, *r* = −0.68 between the *MADS* and *CS*) (Figure 6I). In addition, similar findings were shown between another *AP2* (*LOC_Os03g15660*) TF and the *AK* and *CS* genes separately (Appendix A).

### 2.7. Lysine Content-Related QEI Detection and Candidate Genes

To discover the loci accounted for the potential interactions between the gene and environmental factor, a total of 20 and 30 QEIs were detected in rice grain and leaf lysine content datasets using the 3VmrMLM model (Figure 7A,B, and Appendix A). The PVE by each QEI in grain lysine content datasets ranged from 0.13% to 0.87%, while it in leaf lysine content datasets was from 0.24% to 2.16% (Appendix A). However, no common QEI was detected between the grain and leaf lysine content datasets (Appendix A). In total, 689 and 1066 genes were predicted as the candidate genes of QEIs related to the lysine content in rice grains and leaves (Appendix A). Furthermore, the KEGG pathway analyses showed various genes were involved in the lysine degradation, biosynthesis of amino acids, and glycine, serine, and threonine metabolism pathways in rice grains. Likewise, plenty of genes that participated in the lysine degradation, biosynthesis of amino acids, cysteine and methionine metabolism, and tryptophan metabolism pathways were identified in rice leaves (Table 6). Of these genes, the *LOC_Os01g21380* in the lysine degradation pathway (KEGG annotation: sarcosine oxidase/L-pipecolate oxidase) was a candidate gene (the local LD block: Chr1, 11,942,416–11,956,416 bp) of the QEI-sf0111954416 locus related to grain lysine content in rice (Figure 7C,D, Table 6, and Appendix A). Haplotypic variation analysis showed the grain lysine content of the accessions with Hap1 (GG, *n* = 337) of this gene were significantly higher than those with Hap2 (AA, *n* = 50) in three out of four grain lysine content datasets (Figure 7E–H).

## 3. Discussion

### 3.1. Evaluation of QTNs Associated with Lysine Content in Rice

The number of detected QTNs varied across all the used GWAS models, which resulted from the differences in the genetic algorithm implemented in different models. Even though previous studies suggested that multi-locus models outperform single-locus models on the statistical power of QTN/QTL detection, especially in the accuracy of QTN effect estimation and reduction of false positive rate [35,37,39,52,53,54]. In this study, the largest number of QTNs was detected by GLM across all the lysine content datasets. Additionally, most of the detected common QTNs were identified using the GLM model. In contrast to the averaged *R*^2^ (10.86%) of common QTNs detected by mrMLM-ML models, the averaged *R*^2^ (12.04%) of GLM-detected common QTNs is relatively high. These results are consistent with previous studies suggesting certain advantages of the GLM model on QTN detection [36,55,56,57,58,59,60].

Of note, 14 QTNs detected by GLM were reported in previous study, such as QTN-sf0132487790, QTN-sf0135547034, QTN-sf0141745810, QTN-sf0200277506, QTN-sf0207238898, QTN-sf0315007488, QTN-sf0822844571, QTN-sf0122971223, QTN-sf0135547034, QTN-sf0140365169, QTN-sf0200277506, QTN-sf0207238898, QTN-sf0315007488, QTN-sf0726273868, QTN-sf0810291904, QTN-sf0822844571, QTN-sf1021801564, and QTN-sf1102273126 (Appendix A) [61]. However, few QTNs controlling the lysine content in grains were found in the cereal GWAS study [44]. In the present study, the grain lysine content associated QTN-sf0139799523 and QTN-sf0430630516 were 0.22 kb and 1.38 kb out of the previously reported QTNs [44]. Therefore, adopting multiple statistical models for GWAS may help the identification of both known and novel QTNs associated with the lysine content in rice. Using a similar approach, several novel QTNs associated with leaf free amino acid levels have been identified in rice [37].

### 3.2. Candidate Genes Associated with Lysine Accumulation

To further reveal the genetic basis of lysine accumulation in rice grains and leaves, candidate genes were predicted. Of these genes, *OsAAP3* (*LOC_Os06g36180*), *OsDof3* (*LOC_Os02g1535*), and a *bifunctional aspartokinase*/*homoserine dehydrogenase* gene (*LOC_Os09g12290*) were reported as the lysine biosynthesis and metabolism-related genes, which were also identified in this study (Appendix A) [46,62,63]. Notably, the grain lysine accumulation associated homolog gene *AK* (*LOC_Os07g20544*), encoding a key enzyme in the branch of the Asp family pathway, and lysine is synthesized through this pathway in plants [2,3,64]. The analysis of *AK* haplotype and lysine content of corresponding accessions showed the relatively low grain lysine content of Hap2 accessions (mainly represented for *indica* rice) compared with those Hap1 accessions (mainly enriched in *japonica* rice). Likewise, the Hap2 accessions of the *WRKY* (*LOC_Os12g32250*, the putative TF of *AK*) stood for the *japonica* rice and showed lower grain lysine content than the Hap1 accessions which represented the *indica* rice (Figure 4C–F,I–L). It was identical to the content differences in grain lysine content between *indica* rice and *japonica* rice (Figure 1A). Similar results of *japonica* and *indica* content variation were also obtained in another study about the free branched-chain amino acid (BCAA) content in rice grains [46]. Furthermore, the expression patterns of *WRKY* TF and *AK* gene were all positively correlated, which implied the *WRKY* TF may positively regulate the production of lysine by binding to the cis-elements in the *AK* gene promoter regions in Nipponbare, Minghui63, and Zhenshan97 varieties (Appendix A). In a parallel study, the *OsWRKY78* TF is co-expressed with the branched-chain amino acid (BCAA) content associated gene *OsAUX5* and activates the expression of *OsAUX5* for the BCAA accumulation in rice grains [46]. Moreover, TFs can positively regulate the genes in the metabolite biosynthesis pathways, such as *ZmDOF36* for the starch synthesis in maize [49], *SmMYC2a/b* and *SmMYB98* for the phenolic acid and tanshinone biosynthetic pathway in *Salvia miltiorrhiza* [48], *NbbHLH1* and *NbbHLH2* for nicotine accumulation in *Nicotiana benthamiana* [65].

Additionally, the leaf lysine-associated gene *CS* (*LOC_Os11g33240*) involved in the amino acid biosynthesis pathway was annotated according to the KEGG analysis (K01647). In plants, the citrate synthase (CS) catalyzes the condensation of oxaloacetate (OAA) and acetyl-CoA and further synthesizes the citric acid (CA). Through the production of glutamate, CA can be used for amino acid biosynthesis, such as lysine, proline, and arginine [66,67,68,69]. The *CS* haplotype and lysine content analysis showed the leaf lysine content in Hap2 accessions (mainly *japonica* rice) was higher than those in Hap1 accessions (mainly *indica* rice) across two lysine content datasets (Figure 5B–D). Similarly, the lysine content in Hap2 accessions (mainly *japonica* rice) of the *MYB* (*LOC_Os01g19970*, the putative *TF* of *CS*) was higher than those in Hap1 accessions (mainly *indica* rice) only in one leaf lysine content dataset (Figure 5F–H). It was consistent with the content differences in leaf lysine content between *indica* rice and *japonica* rice (Figure 1C). Similar content alteration between *indica* and *japonica* accessions was also observed in the other studies about the free amino acid content in rice leaves [37,45]. Moreover, the expression relationship of these two genes suggested the *MYB* TF may positively regulate the production of lysine by binding to the cis-elements in the *CS* gene promoter regions in Nipponbare (Appendix A). A previous study reported the *OsbZIP18* TF positively regulates BCAA synthesis by directly binding to cis-elements in the promoters of the biosynthetic genes *OsBCAT1* and *OsBCAT* in rice leaves [45].

The *MADS* (*LOC_Os01g10504*) was potentially able to bind to the promoter regions of the lysine biosynthesis gene *AK* (*LOC_Os07g20544*) and *CS* (*LOC_Os01g19970*). Functional haplotype and content analysis showed a higher lysine content in Hap2 accessions (*indica* rice) than those in Hap1 accessions (*japonica* rice) across all the grain lysine content datasets (Figure 6B–F). It was consistent with the content differences in grain lysine content between *indica* rice and *japonica* rice (Figure 1A). The distinct expression pattern between *MADS* and two potentially target genes, *AK* and *CS*, implied the *MADS* TF plays negatively regulated roles on the lysine accumulation by binding to the cis-elements in the *AK* and CS gene promoter regions in MingHui63 and Zhenshan97 varieties. In plants, multiple genes involved in one or more biosynthetic pathways are negatively/positively regulated by one TF, such as *MYB14* in the sesquiterpenes and flavonoids pathway, *MYC2* in the anthocyanin pathway, and *WRKY76* in the diterpenoid and flavonoid pathway [48,50,51,70,71]. Taken together, the identification of the potential TF targeting the key genes in lysine biosynthesis and metabolism pathways might contribute to the regulation of lysine accumulation in the entire life cycle of rice. To decipher the molecular mechanism of these key genes and corresponding regulators underpinning the lysine accumulation in rice, further validation is warranted to be carried out in the laboratory.

### 3.3. Candidate Gene of Rice Lysine Accumulation Related QEI

Given the challenge of global climate change and the food demands of the ever-growing population, QEI loci accounted for the interactions between the genes and the environment, which hold the potential to be mined for unraveling the genetic basis of complex traits in plant GWAS. Among the candidate genes of QEIs related to the grain lysine content, the genetic variation of the lysine degradation gene *LOC_Os01g21380* (KEGG annotation: sarcosine oxidase/L-pipecolate oxidase) resulted in the lysine content alteration in three out of four content datasets (Figure 7E–H). The lysine content of Hap1 accessions (*indica* rice) was higher than those of Hap2 accessions (*japonica* rice). This result was also in concordance with the content differences between *indica* and *japonica* accessions (Figure 1A). In summary, these suggested this gene might participate in the biological process of lysine accumulation, which was affected by environmental factors. In plants, lysine can be converted to L-pipecolate by the catabolic activity of L-pipecolate oxidase (PIPOX). Importantly, PIPOX is a key enzyme in the lysine metabolism pathway [72,73,74,75,76,77]. Due to the catabolic activity of PIPOX with sarcosine, it is also described as sarcosine oxidase [72,73,74,78]. In this study, the identified QEI loci contain alternative information for the genetic improvement of lysine accumulation to cope with climate change. Moreover, the lysine content of rice accessions can be predicted using these QEI loci in specific environments. Identification of specific genetic markers of QEI loci and their candidate genes will facilitate the development of lysine-rich rice varieties that are better adapted to specific environmental conditions.

### 3.4. Breeding Applications of Lysine Accumulation Associated QTNs and Genes

In this study, the significant correlations between the number of favorable alleles (NFA) and lysine contents (*r* = 0.43~0.71) implied the additive effect of the lysine-accumulation associated QTNs, particularly in the content datasets Grain_env1_r2 and Grain_BLUP (*r* = 0.71) (Appendix A). Based on this, the highest lysine levels were observed in the accessions with a few NFAs, such as W242 with five NFAs in Grain_env1_r1, Grain_env1_r2, and Grain_env2_r2 dataset, C094 with four NFAs in Grain_env2_r1 dataset, W088 with four NFAs in Leaf_env3_r1 dataset, and W001 with four NFAs in Leaf_env3_r2 dataset. These accessions carrying a few NFAs provide potential targets for the lysine-rich rice breeding programs using the loci pyramiding approach. In addition, the detected QTNs are also beneficial for the genomic selection/prediction (GS/GP) breeding programs (predictive ability up to 0.85 in grains and 0.77 in leaves), which may transform the rice nutrient quality breeding from a labor-intensive and time-consuming mode into an efficient and accurate one. The QTNs/QTLs with large and small effects have been successfully applied in the GS breeding to improve the disease resistance, quality, and yield in plants [79,80,81,82,83]. Apart from these QTNs, the identified key genes related to the lysine accumulation in rice grains and leaves can also be applied to the molecular breeding program of lysine-biofortified rice. The higher grain lysine contents were mainly observed in *indica* accessions than in the *japonica* ones (Figure 1A). Therefore, the *indica* accessions with favorable haplotypes of the key genes hold the promise to increase the grain lysine content through the direct hybridization with *japonica* elite varieties, such as the high grain lysine *indica* accession C049 with the favorable haplotype GGCCGGAATTTTGGGGTTAA (Appendix A). In contrast, the *japonica* rice generally showed higher leaf lysine contents than that in the *indica* rice (Figure 1C). Therefore, the leaf lysine level of *indica* rice can be elevated by the hybridization with *japonica* rice, such as the high leaf lysine *japonica* accession W041 with the favorable haplotype GGTTGGAAGGGGCC (Appendix A).

## 4. Materials and Methods

### 4.1. Plant Materials and Sample Sequencing

In this study, a genetic panel containing 387 rice accessions from a previously released worldwide rice collection was used for all the analyses [40]. This diverse panel contains 244 *indica* accessions (*Oryza sativa indica*) and 143 *japonica* accessions (*Oryza sativa japonica*). Of these accessions, 337 accessions are from Asia, followed by 16 accessions from Europe, 14 accessions from South America, 9 accessions from North America, 8 accessions from Africa, and 3 accessions from Oceania. These accessions were planted in the normal rice-growing seasons at two different blocks of Huazhong Agricultural University Experimental Station (Wuhan, China, longitude 114°21′ E, latitude 30°28′ N). The planting density was 16.5 cm between plants in a row, and the rows were 26 cm apart. A randomized complete-block design with two rows of each accession and ten plants in each row was employed in the field-grown plants with two replicates in three consecutive years (2012 for Grain_env1, 2013 for Grain_env2, and 2014 for Leaf_env3). To capture the genetic variation of this plant population, approximately 1 Gb high-quality genome sequences of each accession were obtained through the Illumina HiSeq 2000 genome sequencing platform (Illumina, Inc., San Diego, CA, USA) [40]. The Nipponbare rice reference genome (version MSU 6.1) and its annotation were downloaded from the Rice Genome Annotation Project (http://rice.uga.edu/index.shtml, accessed on 26 December 2023). Using BWA software (v 0.7.17) (https://sourceforge.net/projects/bio-bwa/, accessed on 26 December 2023) with default settings, the clean reads of sequence data were mapped to the MSU 6.1 genome. The SAMtools software (v 1.9) and the HaplotypeCaller, CombineGVCFs, and GenotypeGVCFs functions with default settings in GATK (v 4.0.5.1) (https://gatk.broadinstitute.org/hc/en-us, accessed on 26 December 2023) software were implemented for the SNP joint calling of the 387 rice accessions. A total of 4,630,367 high-quality SNPs were obtained by the filter of -maf 0.05 and -geno 0.1 settings in PLINK software (v 1.9) (https://zzz.bwh.harvard.edu/plink/, accessed on 26 December 2023). These SNPs were used as genotypic datasets in the following analyses.

### 4.2. Metabolite Profiling

In the field, the randomly collected mature grains from three different plants were pooled for further metabolic profiling in the laboratory. The leaves from three random plants at the five-leaf stage were sampled for metabolite extraction as previously described [40,84]. For each accession, two samples of leaf (Leaf_env3_r1 and Leaf_env3_r2 represent 2014_r1 and 2014_r2) and four samples (Grain_env1_r1, Grain_env1_r2, Grain_env2_r1, and Grain_env2_r2 represent 2012_r1, 2012_r2, 2013_r1, and 2013_r2) of grain were prepared for the following metabolomics analyses [84]. For the relative quantification of the free amino acids in the samples above, a liquid chromatography–electrospray ionization–tandem mass spectrometry system was used. Using a mixer mill (MM 400, Retsch GmbH, Haan, Germany) with a zirconia bead for 1.5 min at 30Hz, 100 mg crushed rice sample was extracted overnight at 4 °C with 1.0 mL pure methanol (or 70% aqueous methanol) which contains 0.1 mg/L lidocaine (internal standard) for lipid-solubility free amino acids. A scheduled multiple reaction monitoring method was adopted to conduct the quantification of free amino acids. By dividing the relative signal intensities of metabolites by the intensities of the internal standard (lidocaine, 0.1 mg/L), the relative intensities of free amino acids were normalized. The log_2_-transformed metabolite data were used for improving the normality in further analyses. A metabolic data matrix with the three relative intensities of free amino acid lysine from 2322 runs (387 accessions × six sample sets) was yielded for the rice genetic panel. The broad-sense heritability *H*^2^ of the lysine in rice grains/leaves was estimated using the two and four free lysine content datasets separately. To account for environmental variation, the R package lme4 was implemented to generate the best linear unbiased prediction (BLUP) datasets for the lysine content in rice grains and leaves, respectively [85].

The formula for the estimation of broad-sense heritability *H*^2^:(1)$$H^{2}=\frac{\sigma_{G}^{2}}{\sigma_{G}^{2}+\sigma_{\varepsilon }^{2}}$$
where the $\sigma_{G}^{2}$ is the genotypic variance and $\sigma_{\varepsilon }^{2}$ is the residual variance.

The formula for the calculation of the best linear unbiased prediction (BLUP) value:(2)$$y_{\left(grain\right)}=\mu +Line+Env+\left(Line\times Env\right)+Rep\left(Env\right)+\varepsilon$$
(3)$$y_{\left(leaf\right)}=\mu +Line+Env+\varepsilon$$
where *y*, *μ*, *Line*, and *Env* represent phenotype, intercept, accession effects, and environmental effects, respectively. *Rep* represents different replications, and *ε* represents random effects. *Line* × *Env* is used to display the interaction between accession and environment, and *Rep* (*Env*) indicates the nested effect of replication within the environment.

### 4.3. Population Structure and Linkage Disequilibrium Analysis

To address the redundancy issue of a haplotype block formed by several SNPs within the same linkage disequilibrium (LD) region, using the parameter -indep-pairwise, 200, 100, 0.1 in PLINK software (https://zzz.bwh.harvard.edu/plink/, accessed on 26 December 2023), a total of 107,761 high-quality SNPs were retained for the assessment of genetic relationships. To investigate the population structure of this diverse panel, the principal component analysis (PCA) was implemented by the GCTA software (v1.94.1) based on the high-quality SNPs above (https://github.com/jianyangqt/gcta, accessed on 26 December 2023). Meanwhile, a neighbor-joining (NJ) phylogenetic analysis was performed by the MEGA-CC software (v 11.0.11) with the settings of pairwise gap deletion and 1000 bootstrap replicates [86]. The web tool Inter-active Tree of Life (iTOL) was used for the data visualization of the phylogenetic tree [87]. The ADMIXTURE software (v 1.3.0) was also implemented for the analysis of population stratification [88]. To assess the genome-wide LD decay of this population, the squared correlation coefficient (*r*^2^) between SNPs was calculated using the PopLDdecay software (v 3.42) [89]. The local LD block in a chromosome was estimated by the LDBlockShow software (v 1.40) [90].

### 4.4. Genome-Wide Association Study

Using ten statistical models, the genome-wide association study (GWAS) analyses for lysine content in rice grains and leaves were performed on the genetic panel, including 387 rice accessions with 4,630,367 SNPs and eight content datasets. Of these lysine content datasets, five datasets were the grain lysine content in 2012 and 2013 with two biological replicates (Grain_env1_r1, Grain_env1_r2, Grain_env2_r1, and Grain_env2_r2) and their derived BLUP dataset (Grain_BLUP), and the rest three datasets contained the leaf lysine content in 2014 with two biological replicates (Leaf_env3_r1 and Leaf_env3_r2) and the BLUP dataset of them (Leaf_BLUP). Due to the differences in the genetic algorithm, these models were mainly classified into three groups, namely GLM, MLM-based single-locus models (MLM-SL), and multi-locus random-SNP-effect Mixed Linear Model (mrMLM)-series multi-locus models (mrMLM-ML) for the following identification of common detected QTNs. For instance, MLM-SL contained MLM [30], CMLM [32], and EMMAX [31]. mrMLM-ML included mrMLM [24], FASTmrEMMA [26], FASTmrMLM [91], ISIS EM-BLASSO [25], pKWmEB [27], and pLARmEB [28]. The kinship matrices for individual relationships were generated by each GWAS software package mrMLM (5.0), IIIVmrMLM (1.0). The TASSEL software (v 5.2.40) containing GLM, MLM, and CMLM models was used for the QTN detection with default settings, such as -mlmVarCompEst P3D for MLM and CMLM, -mlmCompressionLevel None for MLM, and -mlmCompressionLevel Optimum for CMLM [92]. The EMMAX software was used to test the marker–trait associations by the implementation of the mixed-model EMMAX with default settings (https://csg.sph.umich.edu/kang/emmax/, accessed on 26 December 2023). The R package mrMLM, including all the mrMLM-ML methods, was implemented to detect the QTNs using parameters SearchRadius = 20, CriLOD = 3, and Bootstrap = FALSE [91]. The R package IIIVmrMLM was used for grain/leaf lysine content-related QEI detection [93]. The parameters for QEI detection were method = Multi_env, SearchRadius = 20, and svpal = 0.01. The marker–trait associations (QTNs/QEIs) in mrMLM and IIIVmrMLM packages were determined by the threshold of LOD score ≥ 3. For the association signals detected by the rest models, the genetic type I error calculator based on the modified Bonferroni correction was adopted to determine the threshold of significant association (*p*-value = 3.22 × 10^−7^ at Type I error α = 0.05 for GLM, and *p*-value = 6.43 × 10^−6^ at α = 1 for MLM, CMLM, and EMMAX). Manhattan plots were generated using the IIIVmrMLM package and R package CMplot with default settings (https://cran.r-project.org/web//packages/CMplot/index.html, accessed on 26 December 2023).

### 4.5. QTN Identification, Candidate Gene Analysis, and Genomic Prediction

To identify the QTNs associated with the lysine content in rice grains and leaves, a GWAS was performed in each grain/leaf lysine content dataset. In each dataset, a common QTN was defined by the QTN, which was detected by two or more GWAS models of GLM, MLM-SL, and mrMLM-ML. The *R*^2^ value of each common QTN was determined by the proportion of total variation explained by the lysine content associated QTN. The rice genes localized within the 122 kb (the averaged whole genome LD decay) flanking regions and the local LD block of a QTN/QEI were potentially predicted as the candidate genes associated with the lysine content in rice grains/leaves. Using the KofamKOALA web tool (https://www.genome.jp/tools/kofamkoala, accessed on 26 December 2023) with default parameters (E-values ≤ 0.01 and hits with scores above the pre-computed adaptive thresholds), the Kyoto Encyclopedia of Genes and Genomes (KEGG) pathway annotation of each candidate gene was obtained. The CandiHap software (v 1.2) was applied to detect the proposed functional haplotypes/sites in the potential candidate genes [94]. Using the multiple comparisons in one-way ANOVA with the LSD method in R package agricolae, the following haplotype and content analysis were conducted for the candidate genes; the different letters indicate statistically significant differences at the 5% probability level. To analyze the transcript factor binding sites of a candidate gene, the function Binding Site Prediction of PlantRegMap web tool (http://plantregmap.gao-lab.org/, accessed on 26 December 2023) was used. The temporal and spatial expression pattern of candidate genes was investigated by the *japonica* rice Nipponbare (http://rice.uga.edu/expression.shtml, accessed on 26 December 2023), *indica* rice Zhenshan97, and Minghui63 RNA-seq data [95].

The R package rrBLUP was implemented to fit the ridge regression best linear unbiased prediction (RR-BLUP) models for the genomic selection/prediction (GS/GP) of lysine content [96,97]. For grain lysine model construction, the grain lysine-associated QTNs and five lysine content datasets (Grain_env1_r1, Grain_env1_r2, Grain_env2_r1, Grain_env2_r2, and Grain_BLUP) were used. Likewise, the leaf lysine-associated QTNs and three leaf lysine content datasets (Leaf_env3_r1, Leaf_env3_r2, and Leaf_BLUP) were used to construct GS/GP models for leaf lysine. Five-fold cross-validation with 500 times was adopted to estimate the predictive ability of each RR-BLUP GS/SP model. The predictive ability (*r*) of each GS/GP model was determined by Pearson’s correlation coefficient between the genomic estimated breeding values (GEBVs) and the observed content values. To investigate the phenotypic variation explained by the SNPs across various lysine content datasets, the SNP-based heritability (*h*^2^) was estimated by the mixed linear model implemented in the GCTA software (v1.94.1) [98].

The formula for the estimation of SNP-based heritability *h*^2^:(4)$$h^{2}=\frac{\sigma_{g}^{2}}{\sigma_{g}^{2}+\sigma_{\varepsilon }^{2}}$$
where the $\sigma_{g}^{2}$ is estimated using the restricted maximum likelihood (REML) method based on the GRM estimated from all SNPs, and $\sigma_{\varepsilon }^{2}$ is the residual variance.

## 5. Conclusions

Using a multi-model GWAS approach, this study identified several QTNs and candidate genes associated with the lysine content in rice grains/leaves and also detected various QEIs and candidate genes related to the grain/leaf lysine content in rice. The reliability and additive effects of 248 and 71 common QTNs associated with grain/leaf lysine content were validated by the significant correlation between the NFA per accession and lysine content (up to 0.71) and the highest accuracy of the GS/GP model (0.85), which provide potential targets for the genetic improvement of lysine accumulation in rice. The three potential regulation modules include positive regulation between the transcription factor *LOC_Os12g32250* and *LOC_Os07g20544* gene in grains, positive regulation between the transcription factor *LOC_Os01g19970* and *LOC_Os11g33240* in leaves, and the negative regulation of the transcription factor *LOC_Os01g10504* to *LOC_Os07g20544* and *LOC_Os01g19970* may hold the promise of the fine-tuning of the lysine accumulation in rice. The 20 and 30 QEIs detected in rice grain/leaf lysine content datasets will facilitate the exploration of gene-by-environment interactions, and ultimately leading to the breeding of lysine-rich and better-adapted rice. Taken together, this study uncovers several novel QTNs and key genes underpinning grain and leaf lysine accumulation and may be expected to provide potential targets for biofortified rice with sufficient levels of lysine and minimal negative effects on plant phenotype.

## Figures and Tables

**Figure 1 ijms-25-04667-f001:**
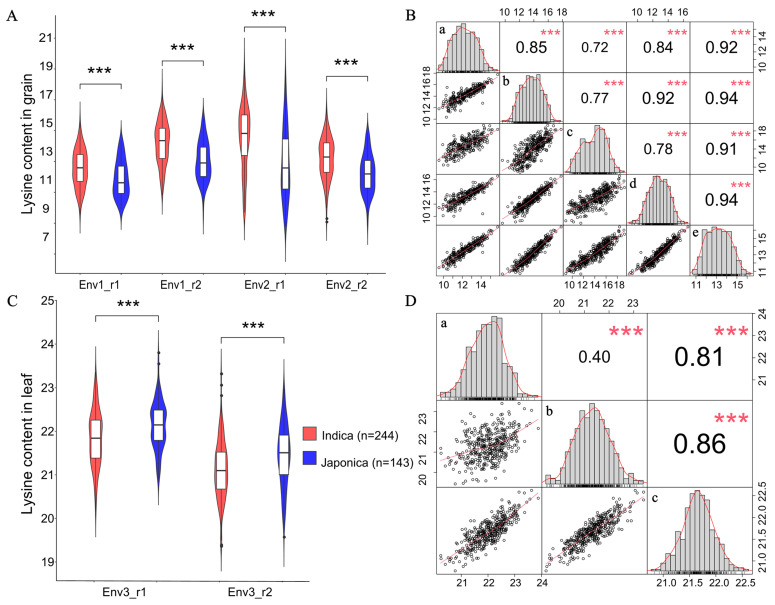
Grain and leaf lysine contents and correlation analyses in rice accessions. (**A**,**C**) Violin plot of lysine content for the 244 *indica* and 143 *japonica* accessions. Env1_r1, Env1_r2, Env2_r1, Env2_r2, Env3_r1, and Env3_r2 represent lysine content datasets with two biological replicates in 2012 and 2013 for grains (Grain_env1 and Grain_env2), and 2014 for leaves (Leaf_env3). (**B**,**D**) Distribution and correlation matrix of lysine content datasets with two biological replicates in Grain_env1, Grain_env2, and Leaf_env3, and the best linear unbiased prediction values (BLUP). For plot (**B**) a–e represent the grain lysine content datasets Grain_env1_r1, Grain_env1_r1, Grain_env2_r1, Grain_env2_r2, and Grain_BLUP. For plot (**D**), a–c represent leaf lysine content datasets Leaf_env3_r1, Leaf_env3_r2, and Leaf_BLUP. *** indicates statistical significance at the 0.1% probability level.

**Figure 2 ijms-25-04667-f002:**
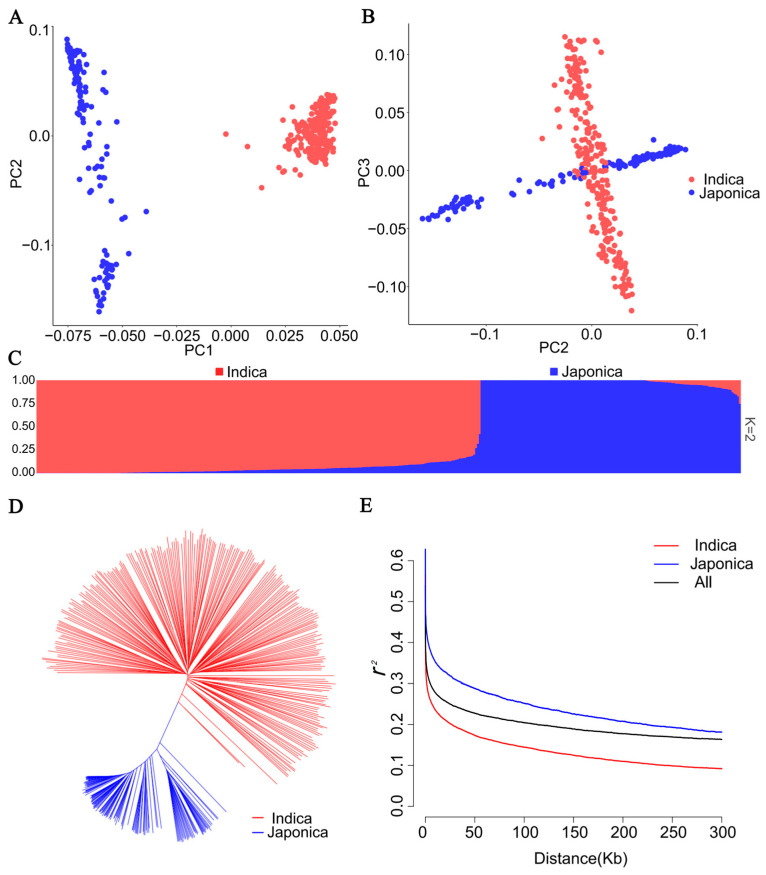
Population structure of 387 rice accessions. (**A**,**B**) Scatter plots of the first three principal components (PCs) of 387 rice accessions. (**C**) Population structure estimated by ADMIXTURE. (**D**) Phylogenetic analysis of 387 rice accessions. (**E**) Genome-wide LD decay analysis of the genetic panel. The squared correlation coefficient (*r*^2^) between SNPs is shown on the y-axis, and the distance of LD decay is shown on the x-axis. The *indica* and *japonica* accessions are indicated in red and blue.

**Figure 3 ijms-25-04667-f003:**
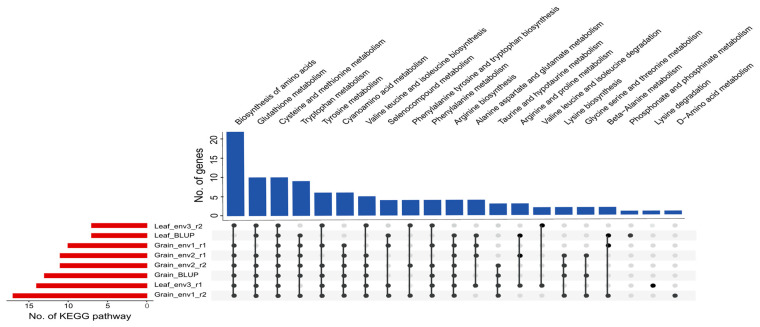
UpSet plot of the candidate genes involved in amino acid-related KEGG pathways. The blue bar chart shows the enriched KEGG pathways of candidate genes associated with lysine accumulation. The red bar chart shows the enriched KEGG pathways of candidate genes detected from different lysine content datasets.

**Figure 4 ijms-25-04667-f004:**
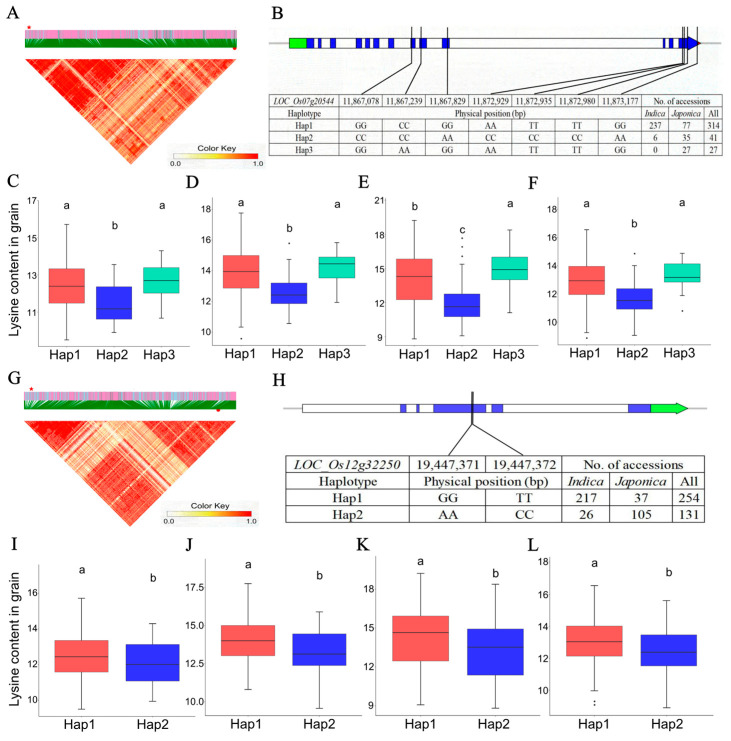
Analyses of the key candidate genes *LOC_Os07g20544* and *LOC_Os12g32250* associated with lysine content in grains. (**A**) Local linkage disequilibrium block analysis, red star and red dot indicate *LOC_Os07g20544* and QTN-sf0711949886 locus. (**B**) Three haplotypes of *LOC_Os07g20544* and their distribution in *indica* and *japonica* accessions. Haplotypic variation and lysine content analysis of *LOC_Os07g20544* in 387 rice accessions in Grain_env1_r1 (**C**), Grain_env1_r2 (**D**), Grain_env2_r1 (**E**), and Grain_env2_r2 (**F**) content datasets. (**G**) Local linkage disequilibrium block analysis, red star and red dot indicate *LOC_Os12g32250* and QTN-sf1219521482 locus. (**H**) Two haplotypes of *LOC_Os12g32250* and their distribution in *indica* and *japonica* accessions. (**I**–**L**): Haplotypic variation and lysine content analysis of *LOC_Os12g32250* in 387 rice accessions in Grain_env1_r1, Grain_env1_r2, Grain_env2_r1, and Grain_env2_r2 content datasets. Different letters indicate statistically significant differences at the 5% probability level in the LSD test. The blue, red, and green boxes represent the coding sequence (CDS), five prime UTR, and three prime UTR of a gene, respectively.

**Figure 5 ijms-25-04667-f005:**
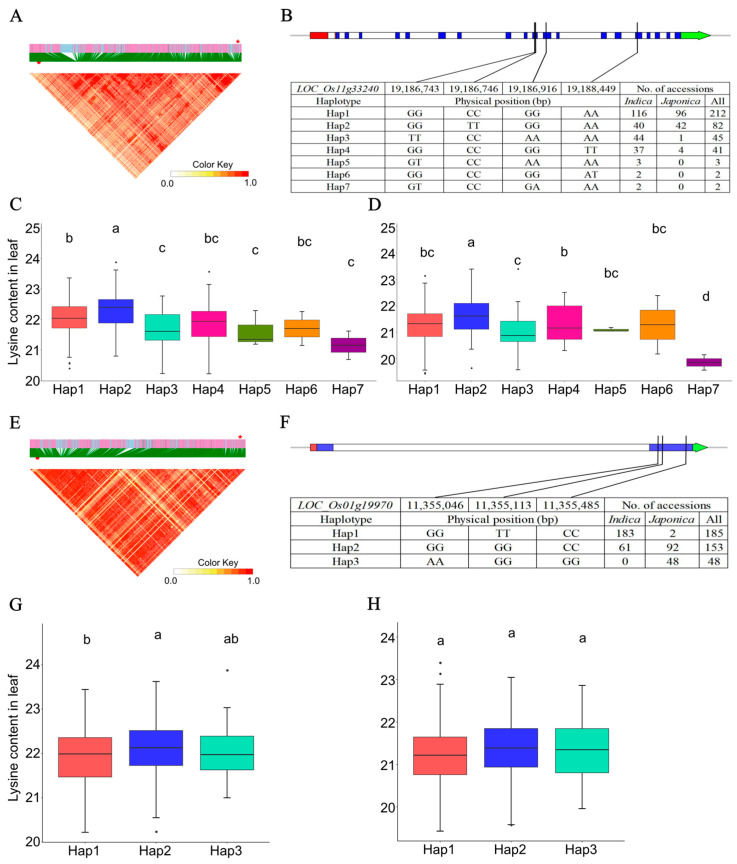
Analyses of the key candidate genes *LOC_Os11g33240* and *LOC_Os01g19970* associated with lysine content in leaves. (**A**) Local linkage disequilibrium block analysis, red star and red dot indicate *LOC_Os11g33240* and QTN-sf1119083279 locus. (**B**) Seven haplotypes of *LOC_Os11g33240* and their distribution in *indica* and *japonica* accessions. Haplotypic variation and lysine content analysis of *LOC_Os11g33240* in 387 rice accessions in Leaf_env3_r1. (**C**) and Leaf_env3_r2 (**D**) content datasets. (**E**) Local linkage disequilibrium block analysis, red star and red dot indicate *LOC_Os01g19970* and QTN-sf0111240543 locus. (**F**) Three haplotypes of *LOC_Os01g19970* and their distribution in *indica* and *japonica* accessions. Haplotypic variation and lysine content analysis of *LOC_Os01g19970* in 387 rice accessions in Leaf_env3_r1 (**G**) and Leaf_env3_r2 (**H**) content datasets. Different letters indicate statistically significant differences at the 5% probability level in the LSD test. The blue, red, and green boxes represent the coding sequence (CDS), five prime UTR, and three prime UTR of a gene, respectively.

**Figure 6 ijms-25-04667-f006:**
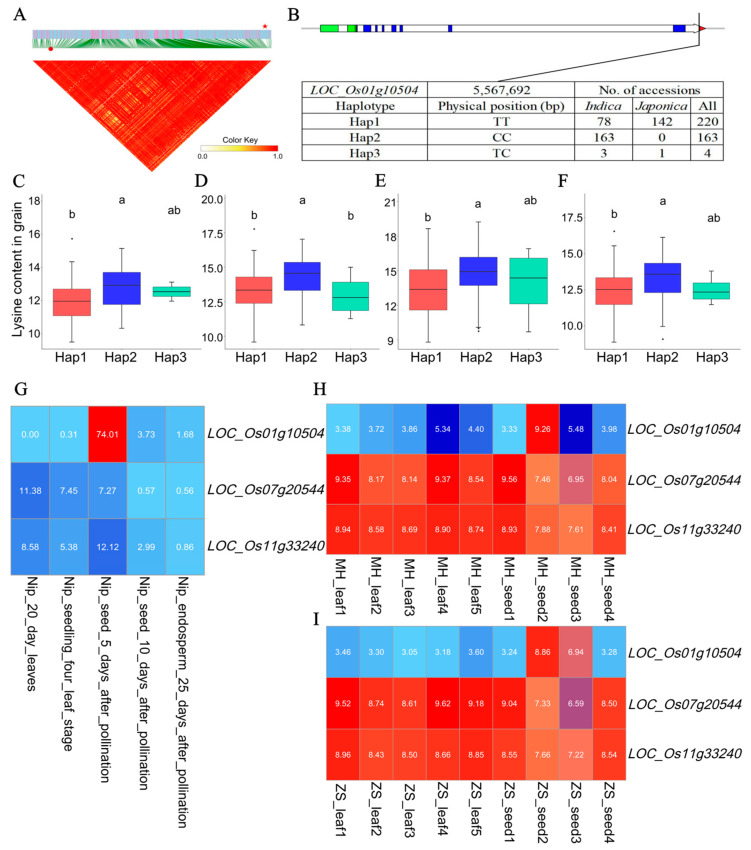
Analyses of the transcription factor *LOC_Os01g10504* related to lysine accumulation in rice. (**A**) Local linkage disequilibrium block analysis, red star and red dot indicate *LOC_Os01g10504* and QTN-sf0105539291 locus. (**B**) Three haplotypes of *LOC_Os01g10504* and their distribution in *indica* and *japonica* accessions. The blue, red, and green boxes represent the coding sequence (CDS), five prime UTR, and three prime UTR of a gene, respectively. Haplotypic variation and lysine content analysis of *LOC_Os01g10504* in 387 rice accessions in Grain_env1_r1 (**C**), Grain_env1_r2 (**D**), Grain_env2_r1 (**E**), and Grain_env2_r2 (**F**) content datasets. Different letters indicate statistically significant differences at the 5% probability level in the LSD test. Heat map of the expression pattern of these key genes in grain and leaf tissue of Nipponbare (**G**), Minghui 63 (**H**), and Zhenshan 97 (**I**) varieties. The red indicates a high expression, and the blue represents a low expression.

**Figure 7 ijms-25-04667-f007:**
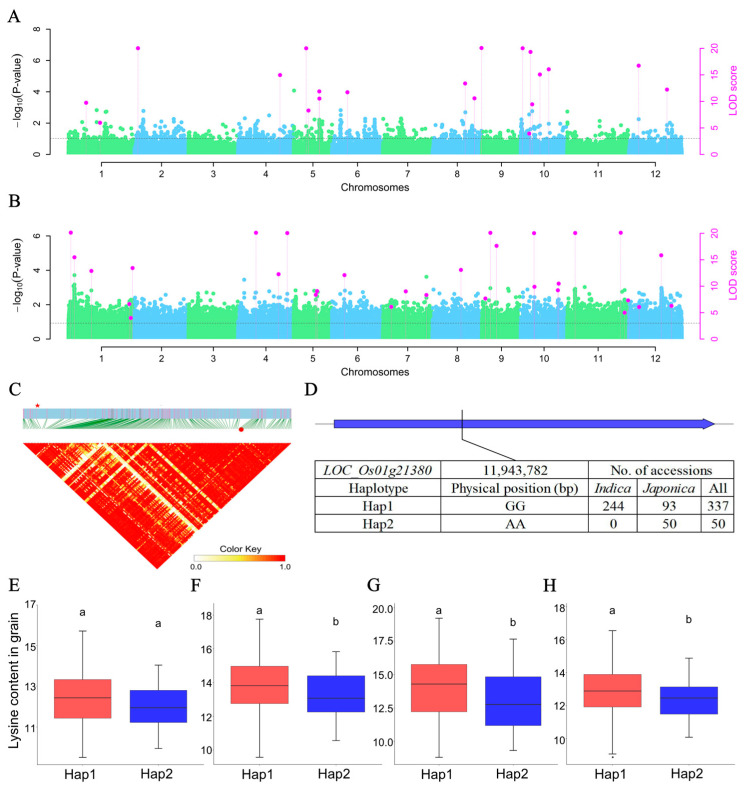
Analyses of lysine accumulation related to QTN-by-environment interactions (QEIs) and the key candidate gene *LOC_Os01g21380*. Manhattan plots for QEIs detected in grain lysine (**A**) and leaf lysine content datasets (**B**). Black horizontal lines in the Manhattan plots represent the genome-wide significant threshold. (**C**) Local linkage disequilibrium block analysis, red star and red dot indicate *LOC_Os01g21380* and QEI-sf0111954416 locus. (**D**) Two haplotypes of *LOC_Os01g21380* and their distribution in *indica* and *japonica* accessions. Haplotypic variation and lysine content analysis of *LOC_Os01g21380* in 387 rice accessions in Grain_env1_r1 (**E**), Grain_env1_r2 (**F**), Grain_env2_r1 (**G**) and Grain_env2_r2 (**H**) content datasets. Different letters indicate statistically significant differences at the 5% probability level in the LSD test. The blue box represents the coding sequence (CDS) of a gene.

**Table 1 ijms-25-04667-t001:** Descriptive statistics of the grain and leaf lysine content datasets.

Dataset	Number	Range	Mean	SD	Variance	Skewness	Kurtosis	CV (%) ^a^	*H* ^2^
Grain_env1_r1	272	6.22	12.35	1.24	1.54	0.10	−0.74	93.59	0.69
Grain_env1_r2	364	8.19	13.75	1.44	2.08	−0.03	−0.63	107.49	
Grain_env2_r1	365	10.41	13.90	2.29	5.26	−0.18	−0.82	165.98	
Grain_env2_r2	365	7.68	12.82	1.38	1.91	−0.09	−0.40	102.84	
Leaf_env3_r1	387	3.65	22.02	0.62	0.39	−0.14	−0.12	43.53	0.16
Leaf_env3_r2	387	3.96	21.31	0.70	0.49	0.07	−0.01	52.62	

^a^ Calculated from the original dataset. CV: coefficient of variation; SD: standard deviation; *H*^2^: broad-sense heritability; Grain_env1_r1, Grain_env1_r2, Grain_env2_r1, Grain_env2_r2, Leaf_env3_r1, and Leaf_env3_r2 represent two replicates in 2012 and 2013 for grains (Grain_env1 and Grain_env2), and 2014 for leaves (Leaf_env3), respectively.

**Table 2 ijms-25-04667-t002:** Common quantitative trait nucleotides (QTNs) detected in grain/leaf lysine content datasets.

Dataset	No. of Detected Common QTNs					*R*^2^ (%)
	GLM|MLM-SL	GLM|mrMLM-ML	MLM-SL|mrMLM-ML	GLM|MLM-SL|mrMLM-ML	Total	
Grain_env1_r1	5	11	4	3	23	0.83–20.25
Grain_env1_r2	21	26	5	3	55	0.12–25.82
Grain_env2_r1	50	12	4	2	68	0.03–24.44
Grain_env2_r2	7	27	3	1	38	0.05–26.08
Grain_BLUP	81	26	4	6	117	0.16–27.65
Leaf_env3_r1	5	17	3	3	28	0.03–16.18
Leaf_env3_r2	16	10	3	1	30	0.27–12.21
Leaf_BLUP	4	11	2	2	19	0.03–13.45

MLM-SL: MLM-based single-locus model, mrMLM-ML: mrMLM-series multi-locus model. Grain_env1_r1, Grain_env1_r2, Grain_env2_r1, Grain_env2_r2, Leaf_env3_r1, and Leaf_env3_r2 represent two replicates in 2012 and 2013 for grains (Grain_env1 and Grain_env2), and 2014 for leaves (Leaf_env3), respectively. BLUP: the best linear unbiased prediction values.

**Table 3 ijms-25-04667-t003:** SNP-based heritability (*h*^2^) and genomic predictive ability (*r*) for the lysine content.

Dataset	*h* ^2^	RRBLUP-*r*
Grain_env1_r1	0.54	0.76
Grain_env1_r2	0.62	0.84
Grain_env2_r1	0.56	0.76
Grain_env2_r2	0.62	0.83
Grain_BLUP	0.64	0.85
Leaf_env3_r1	0.30	0.71
Leaf_env3_r2	0.30	0.65
Leaf_BLUP	0.34	0.77

**Table 4 ijms-25-04667-t004:** Key candidate genes identified for grain lysine accumulation.

Common QTN	Gene Id	KEGG Pathway/Annotation	Functional Annotation	E-Value
QTN-sf0711949886	*LOC_Os07g20544*	Lysine biosynthesis	Aspartokinase	5.9 × 10^−181^
QTN-sf0906935953	*LOC_Os09g12290*	Lysine biosynthesis	Bifunctional aspartokinase/homoserine dehydrogenase	0
QTN-sf0103080436	*LOC_Os01g06600*	Lysine degradation	Glutaryl-CoA dehydrogenase	8.5 × 10^−152^
QTN-sf1012964749	*LOC_Os10g25130*	Alanine, aspartate, and glutamate metabolism	Aminotransferase	6 × 10^−247^
QTN-sf1012964749	*LOC_Os10g25140*	Alanine, aspartate, and glutamate metabolism	Aminotransferase	1.7 × 10^−214^
QTN-sf0311302595	*LOC_Os03g19930*	Alanine, aspartate, and glutamate metabolism	Adenylosuccinate lyase	3.6 × 10^−187^
QTN-sf0717867262	*LOC_Os07g30170*	Beta-Alanine metabolism	Nitrilase	1.4 × 10^−222^
QTN-sf0825353310	*LOC_Os08g40110*	Biosynthesis of amino acids	Peptidase	1.1 × 10^−149^
QTN-sf1013407412	*LOC_Os10g26010*	Biosynthesis of amino acids	Cystathionine gamma-synthase	1.9 × 10^−158^
QTN-sf0419067736	*LOC_Os04g31960*	Biosynthesis of amino acids	Thiamine pyrophosphate enzyme	5.9 × 10^−204^
QTN-sf0419067736	*LOC_Os04g32010*	Biosynthesis of amino acids	Thiamine pyrophosphate enzyme	1.2 × 10^−233^
QTN-sf0100906859	*LOC_Os01g02880*	Biosynthesis of amino acids	Fructose-bisphosphate aldolase isozyme	9.3 × 10^−195^
QTN-sf0110799569	*LOC_Os01g19220*	Cyanoamino acid metabolism	Beta-D-xylosidase	1.7 × 10^−304^
QTN-sf0607725091	*LOC_Os06g13820*	Cysteine and methionine metabolism	Dynamin, putative	0
QTN-sf0803340682	*LOC_Os08g06100*	Tryptophan metabolism	O-methyltransferase	3.5 × 10^−199^
QTN-sf0105539291	*LOC_Os01g10504*	Transcription factor	MADS-box family gene with MIKCc type-box	1 × 10^−95^
QTN-sf0308698430	*LOC_Os03g15660*	Transcription factor	AP2 domain-containing protein	1.7 × 10^−35^
QTN-sf0606188796	*LOC_Os06g11780*	Transcription factor	MYB family transcription factor	4 × 10^−80^
QTN-sf0626549077	*LOC_Os06g44010*	Transcription factor	Superfamily of TFs having WRKY and zinc finger domains	NA
QTN-sf0703936507	*LOC_Os07g07974*	Transcription factor	Tesmin/TSO1-like CXC domain-containing protein	1.9 × 10^−78^
QTN-sf1219521482	*LOC_Os12g32250*	Transcription factor	WRKY DNA-binding domain containing protein	NA
QTN-sf0336203804	*LOC_Os03g64260*	Transcription factor	AP2 domain-containing protein	2.1 × 10^−78^
QTN-sf0101545236	*LOC_Os01g03720*	Transcription factor	MYB family transcription factor	8.7 × 10^−68^

**Table 5 ijms-25-04667-t005:** Key candidate genes identified for leaf lysine accumulation.

Common QTN	Gene Id	KEGG Pathway/Annotation	Functional Annotation	E-Value
QTN-sf0140574604	*LOC_Os01g70220*	Lysine degradation	Histone-lysine N-methyltransferase	1.9 × 10^−121^
QTN-sf1119083279	*LOC_Os11g33240*	Biosynthesis of amino acids	Citrate synthase	2.1 × 10^−140^
QTN-sf0140574604	*LOC_Os01g70170*	Alanine, aspartate, and glutamate metabolism	Transaldolase	2 × 10^−83^
QTN-sf0300274740	*LOC_Os03g01600*	Alanine, aspartate, and glutamate metabolism	Aminotransferase domain-containing protein	3.2 × 10^−147^
QTN-sf0314034319	*LOC_Os03g24460*	Alanine, aspartate, and glutamate metabolism	Aminotransferase domain-containing protein	9 × 10^−57^
QTN-sf0822892970	*LOC_Os08g36320*	Alanine, aspartate, and glutamate metabolism	Decarboxylase	2.5 × 10^−115^
QTN-sf0200325193	*LOC_Os02g01510*	Cysteine and methionine metabolism	Lactate/malate dehydrogenase	2 × 10^−156^
QTN-sf0111240543	*LOC_Os01g19970*	Transcription factor	MYB family transcription factor	1.3 × 10^−76^
QTN-sf0103404473	*LOC_Os01g07120*	Transcription factor	AP2 domain-containing protein	4.5 × 10^−40^
QTN-sf0135366231	*LOC_Os01g60960*	Transcription factor	DUF260 domain-containing protein	NA
QTN-sf0603336542	*LOC_Os06g06900*	Transcription factor	Helix-loop-helix DNA-binding domain-containing protein	NA
QTN-sf0702729577	*LOC_Os07g05720*	Transcription factor	TCP family transcription factor	NA

**Table 6 ijms-25-04667-t006:** QTN-by-environment interactions (QEIs) and candidate genes detected for lysine content in rice grains and leaves.

Dataset	QEI	Gene Id	KEGG Pathway	Functional Annotation	E-Value
Lys_grain	QEI-sf0111954416	*LOC_Os01g21380*	Lysine degradation	FAD-dependent oxidoreductase domain-containing protein	3.2 × 10^−115^
Lys_grain	QEI-sf0519512601	*LOC_Os05g33380*	Biosynthesis of amino acids	Fructose-bisphosphate aldolase isozyme	3.5 × 10^−197^
Lys_grain	QEI-sf1004407883	*LOC_Os10g08022*	Biosynthesis of amino acids	Fructose-bisphosphate aldolase isozyme	9.1 × 10^−196^
Lys_grain	QEI-sf0828052927	*LOC_Os08g44530*	Biosynthesis of amino acids	Dihydroxy-acid dehydratase	2.7 × 10^−301^
Lys_leaf	QEI-sf0103224994	*LOC_Os01g06600*	Lysine degradation	Glutaryl-CoA dehydrogenase	2.5 × 10^−152^
Lys_leaf	QEI-sf1016812592	*LOC_Os10g31950*	Lysine degradation	3-ketoacyl-CoA thiolase	7 × 10^−225^
Lys_leaf	QEI-sf0140517811	*LOC_Os01g70170*	Biosynthesis of amino acids	3-ketoacyl-CoA thiolase	1.1 × 10^−83^
Lys_leaf	QEI-sf1125165035	*LOC_Os11g42510*	Cysteine and methionine metabolism	Tyrosine aminotransferase	1.7 × 10^−166^
Lys_leaf	QEI-sf1220860715	*LOC_Os12g34380*	Cysteine and methionine metabolism	Glutathione synthetase	6.2 × 10^−187^
Lys_leaf	QEI-sf1105428802	*LOC_Os11g10140*	Tryptophan metabolism	Flavin monooxygenase	7.5 × 10^−158^
Lys_leaf	QEI-sf1105428802	*LOC_Os11g10170*	Tryptophan metabolism	Flavin monooxygenase	3.5 × 10^−184^

## Data Availability

All of the phenotypic and genotypic data used in this study are shared in the Appendix A.

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
