# Peer review of "New Insights into the Genetic Basis of Lysine Accumulation in Rice Revealed by Multi-Model GWAS"

_ijms, 2024, doi:10.3390/ijms25094667_

Round 1

Reviewer 1 Report

Comments and Suggestions for Authors

The authors of the submitted study conducted multi-models GWAS of lysine content in 387 rice accessions. They identified common QTNs associated with grain/leaf lysine content and QTNs affected by environment. The authors predicted several critical genes controlling lysine accumulation in rice; the favorable haplotypes can be used in breeding. The authors suggested that multi-model GWAS provides advantages in accuracy in determining the QTNs and reducing false negative rates. Overall, the manuscript is well-written, and the methods are described precisely.

The authors mentioned using LC-MS analysis to determine the amino acid profile of samples. However, it's important to note that the relationship between lysine and total protein content in grains is not straightforward. Research has shown that lysine content can be negatively associated with protein content. The total protein content may decrease as the lysine content increases and vice versa. Do authors observe a positive/negative or neutral trend in the association between lysine and total protein content?

Another minor comment: Please explain the color code in Figure 2 (C and D).

Author Response

Response to Reviewer 1 Comments

Point 1: The authors mentioned using LC-MS analysis to determine the amino acid profile of samples. However, it's important to note that the relationship between lysine and total protein content in grains is not straightforward. Research has shown that lysine content can be negatively associated with protein content. The total protein content may decrease as the lysine content increases and vice versa. Do authors observe a positive/negative or neutral trend in the association between lysine and total protein content?

Response 1: Thank you for the comments. Your question raises an intriguing point for which we may not currently have a satisfactory answer. In the present study, the free amino acid contents were qualified using LC-MS analysis. Due to no available total protein content dataset in hand, we will carry out the association study and related analysis of rice total protein content in future work.

Point 2: Please explain the color code in Figure 2 (C and D).

Response 2: Thank you for the suggestions. We appreciate it for the improvement of this paper. In the updated manuscript, Figure 2 has been replaced to explain color in Figure 2 (C and D). In addition, the sentence “The indica and japonica accessions are indicated in red and blue.” has been highlighted in the legend of Figure 2.

Reviewer 2 Report

Comments and Suggestions for Authors

The Authors performed an extensive and fairly complete GWAS analysis of 387 rice cultivars involving over 4 millions of SNP's to identify  genes and genome regions potentially responsible for the ability of rice to biosynthesize/accumulate lysine at higher or lesser concentrations in leaves and in seeds, generating potentially a very useful information for rice breeders, working on rice biofortification, indicating genes and cultivars with the highest potential to produce transgenic cultivars with improved lysine content in grain protein/s.

Major problem in data interpretation and introduction:

Lysine in human diet comes rather from digestion of lysine containing proteins, than from free lysine in the plant or animal cell.  The article will be more interesting to readers when the authors refer to this, otherwise the readers may assume that free lysine should be present in larger quantities in the leaves (no one knows why?) and in rice seeds.

Minor problems in text:

In abstract – the statement that the rice lacks lysine seems to be a bit too strong, while there is rather a low content of lysine – just a suggestion to reformulate this sentence.

Author Response

Response to Reviewer 2 Comments

Point 1: Lysine in human diet comes rather from digestion of lysine containing proteins, than from free lysine in the plant or animal cell. The article will be more interesting to readers when the authors refer to this, otherwise the readers may assume that free lysine should be present in larger quantities in the leaves (no one knows why?) and in rice seeds.

Response 1: Thank you for the suggestions. We believe this manuscript can be improved with your kind comments. It has been re-edited accordingly in the updated manuscript with track changes.

Point 2: In abstract – the statement that the rice lacks lysine seems to be a bit too strong, while there is rather a low content of lysine – just a suggestion to reformulate this sentence.

Response 2: Thank you for the suggestions. We appreciate it for the improvement of this paper. We fully agree with your suggestion to reformulate this sentence and the statement “rice lacks lysine” has been replaced by “rather a low content of lysine” in the updated manuscript with track changes.

Reviewer 3 Report

Comments and Suggestions for Authors

This exciting work provides insightful information to be implemented in plant breeding programs for crop plant improvement.

Author Response

Response to Reviewer 3 Comments

Point 1: This exciting work provides insightful information to be implemented in plant breeding programs for crop plant improvement.

Response 1: Thank you for the comments. We will continue working on crop plant improvement and hope to make some contributions to this research field and human community.